# Genome-Wide Analysis of the *HSF* Gene Family Reveals Its Role in *Astragalus mongholicus* under Different Light Conditions

**DOI:** 10.3390/biology13040280

**Published:** 2024-04-19

**Authors:** Zhen Wang, Panpan Wang, Jiajun He, Lingyang Kong, Wenwei Zhang, Weili Liu, Xiubo Liu, Wei Ma

**Affiliations:** 1Pharmacy of College, Heilongjiang University of Chinese Medicine, Harbin 150040, China; wz870220@126.com (Z.W.); 15134532248@163.com (P.W.); 17304511090@163.com (J.H.); hljkly970219@163.com (L.K.); liuweili410@126.com (W.L.); 2Experimental Teaching and Practical Training Center, Heilongjiang University of Chinese Medicine, Harbin 150040, China; wenwei0452@163.com; 3College of Jiamusi, Heilongjiang University of Chinese Medicine, Jiamusi 154007, China

**Keywords:** *Astragalus mongholicus*, heat shock transcription factors, gene expression, light treatments, qRT-PCR

## Abstract

**Simple Summary:**

Heat shock (HSF) transcription factors are among the most important transcription factors in plants and are involved in the transcriptional regulation of various stress responses, including drought, salinity, oxidation, osmotic stress, and high light, thereby regulating growth and developmental processes. *Astragalus mongholicus* is a traditional Chinese medicine (TCM) with important medicinal value and is widely used worldwide. Although the *HSF* gene has been reported in most species, it has not been thoroughly studied in *A. mongholicus*. This study not only confirmed all *HSF* genes genome-wide in *A. mongholicus*, but also conducted systematic bioinformatics analysis. At the same time, the expression patterns of *AmHSF* genes in different tissues of *A. mongholicus* and under light response were explored. These results will provide a theoretical basis for understanding the function of *HSF* genes.

**Abstract:**

*Astragalus mongholicus* is a traditional Chinese medicine (TCM) with important medicinal value and is widely used worldwide. Heat shock (HSF) transcription factors are among the most important transcription factors in plants and are involved in the transcriptional regulation of various stress responses, including drought, salinity, oxidation, osmotic stress, and high light, thereby regulating growth and developmental processes. However, the *HFS* gene family has not yet been identified in *A. mongholicus*, and little is known regarding the role of HSF genes in *A. mongholicus*. This study is based on whole genome analysis of *A. mongholicus*, identifying a total of 22 *AmHSF* genes and analyzing their physicochemical properties. Divided into three subgroups based on phylogenetic and gene structural characteristics, including subgroup A (12), subgroup B (9), and subgroup C (1), they are randomly distributed in 8 out of 9 chromosomes of *A. mongholicus*. In addition, transcriptome data and quantitative real time polymerase chain reaction (qRT-PCR) analyses revealed that *AmHSF* was differentially transcribed in different tissues, suggesting that *AmHSF* gene functions may differ. Red and blue light treatment significantly affected the expression of 20 HSF genes in soilless cultivation of *A. mongholicus* seedlings. *AmHSF3*, *AmHSF3*, *AmHSF11*, *AmHSF12,* and *AmHSF14* were upregulated after red light and blue light treatment, and these genes all had light-corresponding *cis*-elements, suggesting that *AmHSF* genes play an important role in the light response of *A. mongholicus*. Although the responses of soilless-cultivated *A. mongholicus* seedlings to red and blue light may not represent the mature stage, our results provide fundamental research for future elucidation of the regulatory mechanisms of *HSF* in the growth and development of *A. mongholicus* and its response to different light conditions.

## 1. Introduction

Crops often encounter abiotic stresses such as high temperature, low temperature, drought, and salinity during growth and development. High temperature stress can block plant growth and development, inhibit physiological activities, and ultimately reduce yield and even cause crop failure [1]. Transcription factors play a crucial role in plant growth. They are key regulators of several signaling networks that activate or repress the transcription of downstream target genes by binding to their promoter or enhancer regions of the corresponding genes, thereby responding to plant growth, development, and environmental stress [2]. The heat shock transcription factor (HSF) is an important transcriptional regulator widely found in plants and is essential for regulating the differential expression of heat shock proteins (HSP) and other functional genes [3]. When plants are subjected to heat stress, it can help plants resist heat stress by binding to the *cis*-element HSE (5′-AGAAG-3′) upstream of the heat stress protein gene and initiating the expression of downstream HSP and its related genes [4]. The HSF protein consists of a DNA-binding domain (DBD), an oligomerization domain (OD or HR-A/B), a nuclear localization signal (NLS), a nuclear export signal (NES), and a C-terminal activation motif (AHA) [5]. The DBD is the most conserved structural domain in *HSFs* and is characterized by a central helix-turn-helix motif that is responsible for binding to *HSE* located in the promoters of target genes [6]. The HR-A/B structural domain is a hydrophobic hepta-peptide repeat sequence. NLS are enriched in Arg (R) and Lys (K) residues, whereas NES are enriched in Leu (L) residues [7]. Several studies have demonstrated that HSFs play important roles in plant responses to high-temperature stress. Overexpression of *ATHSFA1* and *SlHSFA1* significantly increased heat tolerance in *Arabidopsis thaliana* and *Solanum lycopersicum* [8]. The class A HSF gene *TaHsfA2d*, which is similar to *Oryza sativa OsHsfA2d*, was isolated from wheat, and plants overexpressing *TaHsfA2d* in *A. thaliana* showed higher tolerance to high temperatures [9].

Astragali Radix (Huang Qi) is a traditional bulk Chinese medicine. The dried root of the leguminous plant *A. membranaceus* (Fisch.) Bge. and *A. membranaceus* (Fisch.) Bge. var. *mongholicus* (Bge) Hsiao was first recorded in Shennong Ben Cao Jing (Shennong’s Materia Medica) [10]. Huang Qi has anti-inflammatory, antioxidant, and antitumor properties, improves immune function, and protects against cardiovascular and cerebrovascular effects. There are >200 types of traditional Chinese medicines (TCM) made of Huang Qi, which are known as holy medicines for tonifying qi and fixing the surface [11]. To date, various metabolites have been identified in Huang Qi, including saponins, flavonoids, polysaccharides, and amino acids, with isoflavones and saponins being the main bioactive components [12]. In recent years, there has been increasing evidence of the therapeutic activity of *Astragalus* isoflavonoid constituents, and there has been great interest in improving the yield and quality of *A. mongholicus* [13]. Previous studies have shown that many biotic and abiotic factors, such as soil microorganisms, ultraviolet light, drought, and temperature, are closely related to the yield of *Astragalus* isoflavones [14]. Under drought stress, the combined action of six bacterial consortia significantly promotes the accumulation of isoflavones in *A. mongholicus*, promotes plant growth, and increases the antioxidant enzyme activities of *A. mongholicus* to alleviate the adverse effects caused by drought [15]. Under UV-B radiation, calycosin and calycosin-7-glucoside in *A. membranaceus* and *A. mongholicus* accumulate in large quantities to help plants respond to UV-B stress [16]. Overall, studies on the effect of *Astragalus* isoflavone synthesis have focused on different biotic and abiotic stresses, whereas the effect of light treatment at a single wavelength for a short period on the isoflavone content of *A. mongholicus* at the seedling stage in soilless cultures is not known. Recently, whole-genome sequencing of *A. mongholicus* was completed and the biosynthetic pathway of isoflavones was identified [17]. However, little research has been conducted to develop high-quality *A. mongholicus* varieties with high isoflavone content by studying the molecular mechanisms of light response.

Several plant HSF TFs have been extensively studied for their key regulatory functions in response to development and different stresses, and previous studies have shown that not only high temperature but also different light, low temperature, and salinity–drought stresses affect the expression of HSF genes [18]. Thus, the molecular regulation of *HSFs* in response to various stressors is a complex and delicate process that involves changes in multiple genes and metabolites. Although *A. mongholicus* has a long medicinal history and its isoflavone biosynthetic pathway has been elucidated, HSF TFs have not been thoroughly studied. Therefore, we used a bioinformatics approach to identify HSF gene family members under the whole *A. mongholicus* genome and in this study examined the physical and chemical properties, phylogenetics, gene structure, collinear relationship, and differential expression in different tissues of *A. mongholicus*. In addition, the expression levels of *AmHSF* genes under different light conditions were analyzed by qRT-PCR to provide reference data for further study of *AmHSF* gene function. It is important to note that light-treated soilless-cultivated seedlings of *A. mongholicus* were selected for this study, and alterations at this stage in response to light and *HSF* gene expression may not represent a response to the mature stage. Taken together, these findings provide data support for future in-depth studies on the function of *AmHSF* genes and subsequently lay the foundation for comprehensively revealing the role of the HSF gene family in the growth and light response.

## 2. Materials and Methods

### 2.1. Soilless Cultivation of A. mongholicus Culture and Light Treatment

*A. mongholicus* seeds purchased from the Anguo market were used as experimental materials and were identified by Ma Wei from Heilongjiang University of Traditional Chinese Medicine. Fully mature seeds with full grains were selected, washed first, and then soaked for 8–10 h, and the clean seeds were placed in a soilless cultivation device developed by our research group for cultivation (patent number: 201220490816.3). The plants were incubated in a light incubator with an average light intensity of 22,000 l×, a day/night temperature of 25/20 °C, and a light/dark photoperiod of 16/8 h. The nutrient solution used was Hoagland’s nutrient solution (Coolaber Science & Technology, Beijing, China), which was changed every 2 weeks. When the age of the seedlings was 40 d, the roots, stems, and leaves were sampled from the plants with good growth and consistency, and the samples were frozen in liquid nitrogen and stored in the refrigerator at −80 °C.

### 2.2. Identification and Physicochemical Properties Analysis of the HSF Gene Family in A. mongholicus

*A. mongholicus* genome was downloaded from the Global Pharmacopoeia Genome Database (GPGD; http://www.gpgenome.com/, 2 February 2024). All CDS sequences in the genome of *A. mongholicus* were extracted and converted into protein sequences using TBtools software V2.083 [19]. The hidden Markov model (PF00447) of the *HSF* gene was downloaded from the Pfam database (http://pfam.xfam.org/, accessed on 2 February 2024) to screen for the *HSF* gene in the genome of *A. mongholicus* with an E value of 10^−10^. The candidate sequences were validated using the National Center of Biotechnology Information (NCBI)-CDD (https://www-ncbi-nlm-nih-gov-443.webvpn.nefu.edu.cn/, accessed on 2 February 2024), Plant Transcription Factor Database (TFDB; https://planttfdb.gao-lab.org/index.php, accessed on 2 February 2024), and Simple Modular Architecture Research Tool (SMART; http://smart.embl.de/, accessed on 2 February 2024). After eliminating genes that did not contain DBD conserved domains and redundant genes, they were summarized and named based on their sequential order of chromosome position. Physicochemical properties such as amino acid number, molecular weight, and hydrophobicity of the AmHSF protein were analyzed using the online software Expasy-ProtParam (https://web.expasy.org/protparam/, accessed on 2 February 2024).

### 2.3. Phylogenetic Analysis of AmHSF Proteins

The *A. thaliana* HSF protein sequence was downloaded from The Arabidopsis Information Resource (TAIR) database (http://www.arabidopsis.org/, accessed on 4 February 2024). Multiple sequence comparisons of AtHSF and AmHSF protein sequences were performed using the MEGA-X software [20], and a phylogenetic tree was constructed based on the neighbor-joining (NJ) method, with the calibration parameter bootstrap set to 1000. Phylogenetic trees were constructed using the EvolView online tool (https://evolgenius.info//evolview-v2/login, accessed on 4 February 2024). Members of the AmHSF family were grouped into pre-existing subgroups in *A. thaliana* for cluster analysis.

### 2.4. Conserved Motifs and Gene Structure Analysis of the AmHSF Gene Family

The AmHSF protein sequence was subjected to motif analysis using the MEME online server (https://meme-suite.org/meme/tools/meme, accessed on 5 February 2024) with 10 predicted motifs. In addition, the identified *AmHSF* gene sequences were processed and screened in their annotation files based on the identified sequences. The results were processed using the online software Gene Structure Display Server 2.0 (GSDS; http://gsds.gao-lab.org/, accessed on 5 February 2024) to complete the gene structure analysis and visualized using TBtools software.

### 2.5. Analysis of Cis-Elements of the AmHSF Gene Family

The 2000 bp sequence upstream of the *HSF* gene was obtained from the genome of *A. mongholicus* using TBtools software, and the *cis*-elements were predicted using the online website PlantCARE (http://bioinformatics.psb.ugent.be/webtools/plantcare/html, accessed on 6 February 2024), which used the TBtools software to map and analyze the main *cis*-elements.

### 2.6. Chromosomal Mapping and Collinearity Analysis of AmHSF Gene

TBtools software was used to perform chromosome localization and visualization of the AmHSF gene family. The whole gene data of *Cannabis sativa* (GCA_029168945.1), *Malus domestica* (GCF_002114115.1), and *Solanum lycopersicum* (GCA_000188115.4) were downloaded from the NCBI website and analyzed for inter-species covariance using MCScanX software [21], visualized using TBtools software, and the *HSF* genes were highlighted.

### 2.7. Analysis of AmHSF Gene Family Expression Pattern in Different Tissue Sites and Quantitative Reverse Transcription Polymerase Chain Reaction (qRT-PCR) Validation

RNA sequencing (RNA-seq) data of *A. mongholicus* soilless-cultivated seedlings in different tissue sites were sequenced by our group. Fragments per kilobase of transcript per million fragments mapped (FPKM) were used as a measure of gene expression abundance, and TBtools software was used to analyze the expression patterns of the HSF gene family in roots, stems, and leaves. The log_2_ (FPKM + 1) transformation was used for processing.

Total RNA was extracted using the RNAprep Pure Plant kit (Tiangen Biotech, Beijing, China), reverse transcribed according to the MS 1st Strand cDNA Synthesis SuperMix for qPCR (+gDNA wiper) kit (Msunflowers Biotech Co., Ltd., Beijing, China), and analyzed using qRT-PCR on the AriaMx Real-Time PCR System platform using the reaction system configured using the ChamQ Universal SYBR qPCR Master Mix kit (Vazyme Biotech Co., Ltd., Nanjing, China). The *18s* gene was used as the reference gene, and the amplification procedure was as follows: pre-denaturation at 95 °C for 5 min, followed by 40 cycles at 95 °C for 10 s and 60 °C for 30 s. Quantitative data were analyzed using the 2^−∆∆CT^ method [22], and *t*-tests were performed on the results using GraphPad Prism 8.0.2 software.

## 3. Results

### 3.1. Identification and Physicochemical Properties Analysis of the HSF Gene Family in A. mongholicus

Based on HMM, a comprehensive search was conducted for the *HSF* genes of *A. mongholicus*. After removing the incomplete domains and redundant sequences, 22 *HSF* genes were identified in the entire genome of *A. mongholicus* (Appendix A). Renaming them as *AmHSF1*-*AmHSF22* based on their position on the chromosome, with the prefix Am indicating *A. mongholicus*, analysis of the physicochemical properties showed that the amino acid length of the AmHSF proteins ranged from 204 aa (AmHSF13) to 496 aa (AmHSF10). Their molecular weights ranged from 23747.23 Da to 54897.07 Da, and their isoelectric points ranged from 4.83 (AmHSF16) to 8.73 (AmHSF9). The grand average hydropathicity (GRAVY) of the AmHSF proteins was less than 0, indicating that the AmHSF proteins were all hydrophilic.

### 3.2. Chromosomal Distribution and Gene Duplication Event Analysis of AmHSF Genes

The identified 22 *AmHSF* genes were unevenly distributed on the 8 chromosomes of *A. mongholicus*. We named these sequences *AmHSF1*-*AmHSF22* based on their specific chromosomal locations. Except for chromosome 8, each chromosome contained a different number of *AmHSF* genes. The distribution of *AmHSF* genes on chromosome 5 was the highest (five), followed by chromosomes 2 and 4, each containing four genes. The distribution on chromosome 1 was the lowest, with only one *AmHSF* gene (Figure 1).

One of the important factors leading to the functional differentiation and amplification of genes is gene duplication. Of the 22 *AmHSF* genes, seven pairs of fragment duplication events were identified, with most duplication events on chromosome 4, and only one pair on the remaining chromosomes. These results suggest that these fragment duplication events were primarily responsible for the evolution of *AmHsf* genes.

### 3.3. Phylogenetic Analysis of AmHSF Gene Family

To further clarify the family taxonomic relationship of HSF in *A. mongholicus*, we constructed a phylogenetic tree by combining the HSF proteins in *A. thaliana* with the HSF proteins in *A. mongholicus* (Appendix A) (Figure 2). The results showed that the 22 AmHSF proteins could be categorized into three subfamilies: Group A, Group B, and Group C. Group A contained eight subfamilies (A1, A2, A3, A4, A5, A6, A8, and A9), group B contained three subfamilies (B1, B2, and B3), and Group C contained only one C1 subfamily. Notably, subfamilies A7, B4, B5, and C2 were missing in both *A. thaliana* and *A. mongholicus*, suggesting evolutionary specificity among the species. In addition, different subfamilies in families A and B contained different numbers of family members, such as A1 (AmHSF10 and AmHSF12), A4 (AmHSF14 and AmHSF17), A5 (AmHSF2 and AmHSF15), and A6 (AmHSF5 and AmHSF22), all of which contained two family members, whereas the remaining A families only had one gene. From a genetic perspective, there were 10 pairs of directly homologous genes between *A. thaliana* and *A. mongholicus*, including *AmHSF8*, *AtHSFB2B*, *AmHSF21,* and *AtHSFB3*, indicating that their evolutionary trends are consistent and may have similar biological functions.

### 3.4. Analysis of Conserved Motifs and Gene Structures in AmHSF

The conserved motifs and structural features of the AmHSF genes were analyzed. To understand the potential functions of *AmHSF* genes, the conserved motifs of AmHSF proteins were predicted using the Motif Elicitation website. Ten conserved motifs in the AmHSF proteins were predicted, and a motif corresponding to the conserved domain was identified (Figure 3A). Based on the conserved amino acid sequences of motifs 1, 2, 3, 5, 7, and 9, these were divided into five categories (DBD, HR-A/B or OD, NLS, NES, and AHA). Gene structure analysis showed that the *AmHSF* gene had little difference in its structural features (Figure 3B). *AmHSF* genes contained between 0 and 3 introns, 17 genes contained 1 intron, 4 genes contained 2 introns, and *AmHSF*2 contained the most introns (3).

### 3.5. Syntenic Analysis of AmHSF Genes in Different Species

To further understand the evolutionary pattern of the AmHSF gene family, we conducted a genomic association analysis between *A. mongholicus* and four different dicotyledonous plants (*Malus domestica*, *Solanum lycopersicum*, *Cannabis sativa*, and *Cicer arietinum*), among which *C. arietinum* and *A. mongholicus* are both leguminous species (Figure 4). The results showed that *A. mongholicus* had the strongest collinearity with *M. domestica* with 41 gene pairs, followed by *S. lycopersicum* with 28 gene pairs, *C. arietinum* with 26 gene pairs, and *C. sativa* with the worst collinearity, with only 10 gene pairs. In addition, the *HSF* genes that were collinear with these four species in *A. mongholicus* were *AmHSF1*, *AmHSF5*, *AmHSF6*, *AmHSF9*, *AmHF20*, and *AmHSF22*, indicating that these six *AmHSF* genes may have existed before the differentiation of these species.

### 3.6. Cis-Element Analysis of AmHsf Genes

Using the online tool plantCARE, 18 types of 422 valuable *cis*-elements were identified in the promoter region of *AmHsf* gene after screening (Figure 5). These *cis*-elements can be broadly classified into the following three categories: hormonal responses, environmental stress, and growth and development. The diversity of these *cis*-elements suggests that the different *AmHSF* genes have different functions. All *AmHSF* genes had light *cis*-elements, except for *AmHSF17*. All other *AmHSF* genes had abscisic acid-responsive elements. In addition, low-temperature *cis*-elements, anaerobic induction elements, and defense and stress response elements were present in the promoter region of the *AmHSF* gene, indicating that they are involved in different stress responses. Notably, the *AmHSF* genes promoter region was also a *cis*-element in the MYB transcription factor-binding site for light response, drought response, and flavonoid synthesis, suggesting that *AmHSF* and *AmMYB* may form a molecular regulatory network that jointly regulates the growth and development of *A. mongholicus*. In summary, these results indicated that the *AmHSF* gene responds to a wide range of hormones and can cope with various stresses.

### 3.7. Expression Profile and qRT-PCR Verification of AmHsf Genes in Different Tissues of A. mongholicus

To explore the potential functions of *the AmHsf* genes during the development of different tissues and organs, RNA-seq data from the roots, stems, and leaves of *A. mongholicus* were used to examine their expression patterns (Appendix A) (Figure 6). Thirteen genes were expressed in all tissues of *A. mongholicus* (FPKM > 0.5), of which *AmHSF8*, *AmHSF10*, *AmHSF19,* and *AmHSF20* were the most highly expressed (FPKM > 10). Seven genes had extremely high expression levels in roots, five genes had extremely high expression levels in stems, and four genes had extremely high expression levels in leaves. *AmHSF3* was specifically expressed in leaves, and *AmHSF21* and *AmHSF22* were specifically expressed in roots. *AmHSF7*, *AmHSF11*, *AmHSF13,* and *AmHSF17* were not expressed in all tissues; these genes may be false and may be expressed in response to stimulation or special time and space. The different expression patterns of *AmHsf* genes have different functions in the growth and development of *A. mongholicus*.

To confirm the accuracy of the RNA-seq data and further validate the expression patterns of key *AmHSF* genes in different tissues of *A. mongholicus*, 11 *AmHSF* genes with high expression levels were selected for qRT-PCR analysis (Figure 7). The results showed the expression patterns of nine genes in the roots, stems, and leaves of *A. mongholicus* together with RNA-seq data. Notably, the expression of *AmHSF10* and *AmHSF15* was the highest in the roots in the RNA-seq data, whereas the qRT-PCR results showed that their expression was higher in the stems.

### 3.8. Analysis of AmHSF Expression Patterns after Different Light Treatments

Based on the prediction of *cis*-elements in the *AmHSF* promoter, all *AmHSF* genes had light-responsive *cis*-elements. Therefore, we compared the gene expression changes of *AmHSF* after light treatment to determine whether the expression of *HSF* genes would be affected during light treatment (Appendix A). We subjected A. mongholicus to dark, red, and blue light treatments and performed qRT-PCR on the responses of 22 *AmHSF* genes to assess their expression patterns and further understand their possible functions under different light treatments. The results showed that blue and red light treatments significantly altered the expression of these genes (Figure 8). The results showed that the blue and red light treatments significantly altered the expression of these genes. All *AmHSF* genes, except *AmHSF11* and *AmHSF21*, showed higher responses to red light than to blue light. Seventeen genes were upregulated by red light-induced expression, five genes were upregulated by blue light-induced expression, and *AmHSF3*, *AmHSF11*, *AmHSF12*, and *AmHSF14* were upregulated under both red and blue light treatments. Four genes were previously not expressed in various tissues, and *AmHSF13* and *AmHSF17* were still not expressed, whereas *AmHSF7* and *AmHSF11* were expressed under light treatment, indicating that blue and red light can activate the expression of these genes.

## 4. Discussion

As plants are stationary, they have evolved complex transcriptional regulation systems to cope with changes in their surroundings [23]. Studies have shown that the *HSF* gene is a heat shock transcription factor that plays an important role in plant growth, development, and response to abiotic stress [24]. HSF TFs have been identified and studied in a variety of plants. The number of HSF TFs is not the same in different species, e.g., 29 in *Fagopyrum tataricumt* [25], 21 in *A. thaliana* [26], 19 in *S. lycopersicum*, 16 in *Medicago sativa* [27], and 13 in *Beta vulgaris* [28]. However, the HSF family has not been systematically studied in *A. mongholicus*. In the present study, we identified 22 *HSF* genes in the genome of *A. mongholicus,* which provided evidence to explore the evolutionary relationships of *AmHSF* genes and identify key genes related to plant developmental processes and responses to different light treatments. Based on the reported phylogenetic tree results for *A. thaliana HSF* genes, we divided them into three groups: A, B, and C. The number of HSF genes in *A. mongholicus* is roughly equivalent to that in *A. thaliana*. There are more *AmHSF* genes in the A5, A6, B1, B4, and B2 subfamilies; however, they are not found in the A7 subfamily. The reasons for the increase or decrease in these genes require further research but are typically related to the expansion or contraction of gene families during evolution [29].

In addition, we determined that the gene structure of *AmHSF* has a similar exon–intron pattern (Figure 3). The number of introns can affect gene expression efficiency [30], and the presence of similar domains and conserved motifs suggests that they may have similar functions. Not only are the gene structures similar, but each subgroup of *AmHSF* has a similar motif. The three helical bundle structures of the DBD domain are the most conserved, and motifs 1, 2, and 3, which encode the DBD domain, are present in all *AmHSF* subgroups. The amino acids encoded by motif8 are LFGV-peptides that regulate the response of HSF genes to heat shock and belong to the B subfamily. The HSF members in each subgroup have similar motifs, which have also been reported in *Gossypium hirsutum* [31] and *Populus trichocarpa* [32].

The function of AmHSFs can be predicted by combining transcriptome data, qRT-PCR, and phylogenetic analyses of *A. thaliana* homologous genes [33]. According to the transcriptome data analysis of different tissues of *A. mongholicus*, the *AmHSF* gene family has tissue-specific expression levels (Figure 6 and Figure 7). Owing to the unique structure of the B subfamily, most members of the HSF can control the development of nutrition and reproductive tissues. For example, *AtHSFB2A* plays an important role in regulating gametophyte development in *A. thaliana* [34]. According to the phylogenetic relationships (Figure 2), *AmHSF1*, *AmHSF4*, *AmHSF8*, *AmHSF9*, *AmHSF19*, and *AmHSF21* belong to the B subfamily and are highly expressed in the roots, which may regulate root development. It is worth noting that *AtHSFB4* can regulate the root development of *A. thaliana* [35], but its homologs *AmHSF7* and *AmHSF11* are not expressed in the roots, which may require further research. Similarly, some genes in subgroup A have been reported to regulate plant developmental processes. *AtHSFA9* from the A9 subgroup has been reported to be involved in the regulation of *A. thaliana* seed development [36], and its homologous *AmHSF18* may have the same biological function.

The *HSF* family is particularly important for the stress response [37]. Overexpression of *AtHSFA6A* increases drought tolerance in *A. thaliana* [38], as does the overexpression of the *Glycine max* GmHSF34 gene in *A. thaliana*. *GmHSFB2b* can promote the synthesis of flavonoids, reduces ROS accumulation in vivo, and improves the salt tolerance of *G. max* [39]. In *A. mongholicus*, HSF has multiple stress-related *cis*-elements in their promoter regions (Figure 5). *AmHSF5* and *AmHSF22*, homologous to *AtHSFA6A,* and *AmHSF6*, homologous to *GmHSF34*, contain *cis*-elements for a variety of abiotic stresses, including anaerobic induction, stress resistance, and participation in drought induction. *AmHSF4*, *AmHSF5*, *AmHSF6*, *AmHSF10*, *AmHSF14*, *AmHSF16*, *AmHSF17*, *AmHSF20*, and *AmHSF21* have low-temperature-induced response elements that may help *A. mongholicus* resist cold stress. Similar studies have been conducted on *A. thaliana*, *P. trichocarpa*, and *Vigna radiata* [40]. In summary, genes with stress-responsive *cis*-elements may serve as candidates for abiotic stress resistance.

The *HSF* family not only responds to abiotic stress, but also plays an important role in light response. Studies have shown that blue light and red light significantly enhance the expression of *HSF* genes in *C. sativa* and affect cannabinoid biosynthesis [41]. In *A. thaliana*, *HSFA1s* is a key regulator that integrates light and temperature signals. The *HSFA1s* gene interacts with the PIF4 gene to form a module that mediates the regulation of daytime thermal morphogenesis, which helps the plant adapt better to summer [42]. Overexpression of *HSFA2* protects *A. thaliana* against high light stress [43]. Our data showed that blue and red light treatments altered *AmHSF* expression in soilless-cultivated seedlings of *A. mongholicus* (Figure 8). We performed qRT-PCR analysis of 22 *AmHSF* genes to analyze their expression patterns of *AmHSF* genes under red and blue light treatments. Seventeen *AmHSF* genes were upregulated under red light, and five genes were upregulated under blue light. Among these, *AmHSF3*, *AmHSF11*, *AmHSF12,* and *AmHSF14* were upregulated under both red and blue light. In addition, we showed that all these upregulated genes possessed light-responsive *cis*-elements. It has been reported that red and blue light can promote the accumulation of *Astragalus* isoflavones [44]. In our study, red light and blue light can also positively regulate the expression of *AmHSF* genes, so it is speculated that *AmHSF* genes may have the function of regulating the accumulation of isoflavones in *A. mongholicus* under different light, but its true biological function needs to be further verified. However, soilless-cultivated *A. mongholicus* seedlings may not represent a more realistic stage of maturity for light treatment of *AmHSF* expression. In view of the above facts, further studies are needed to understand its effects on mature *A. mongholicus*. By studying the effects of light on *AmHS* expression, we hope to contribute to a better understanding of the molecular mechanisms underlying isoflavone synthesis in *A. mongholicus*.

## 5. Conclusions

In this study, 22 *AmHSF* gene family members were identified in the whole genome of *A. mongholicus*, and their physicochemical properties, chromosome distribution, phylogeny, gene structure, collinearity, and *cis*-elements were systematically analyzed. Transcriptome analysis and examination of gene expression patterns in different tissues revealed that *AmHSF* genes were specifically expressed in different tissues. In addition, the expression of *AmHSF* genes was significantly altered by blue and red light treatment. In summary, this is a preliminary study of the *A. mongholicus HSF* gene family that provides a reference for further studies of the function of *HSF* genes in different species and expands our understanding of the molecular mechanism of *A. mongholicus* in response to different light.

## Figures and Tables

**Figure 1 biology-13-00280-f001:**
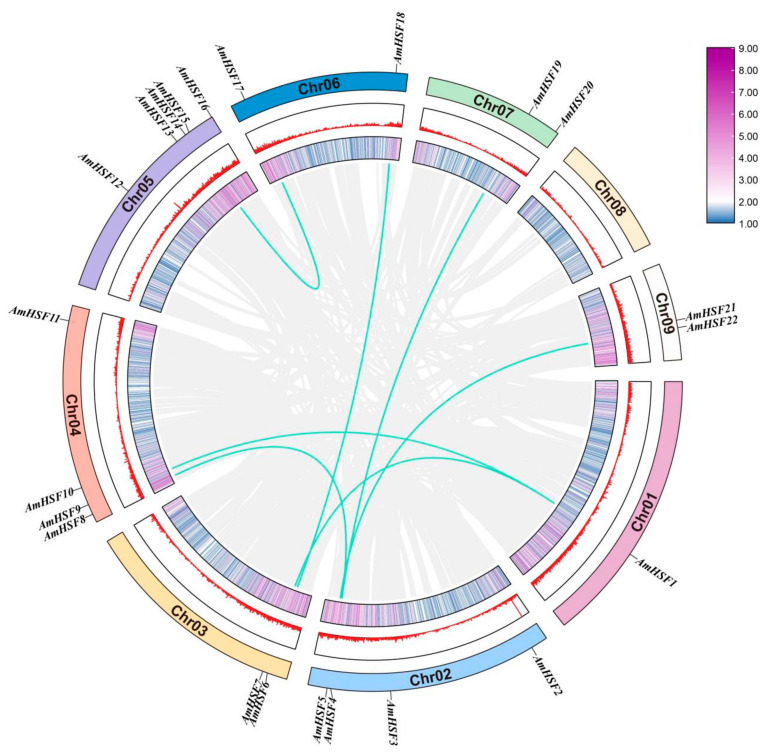
Chromosomal distribution and duplication events. Highlighted lines indicate gene duplication events in the *AmHSF* gene, and gray lines show collinear pairs of all *A. mongholicus* genes. Red line, Purple lines and the blue bar indicate the gene density in each chromosome.

**Figure 2 biology-13-00280-f002:**
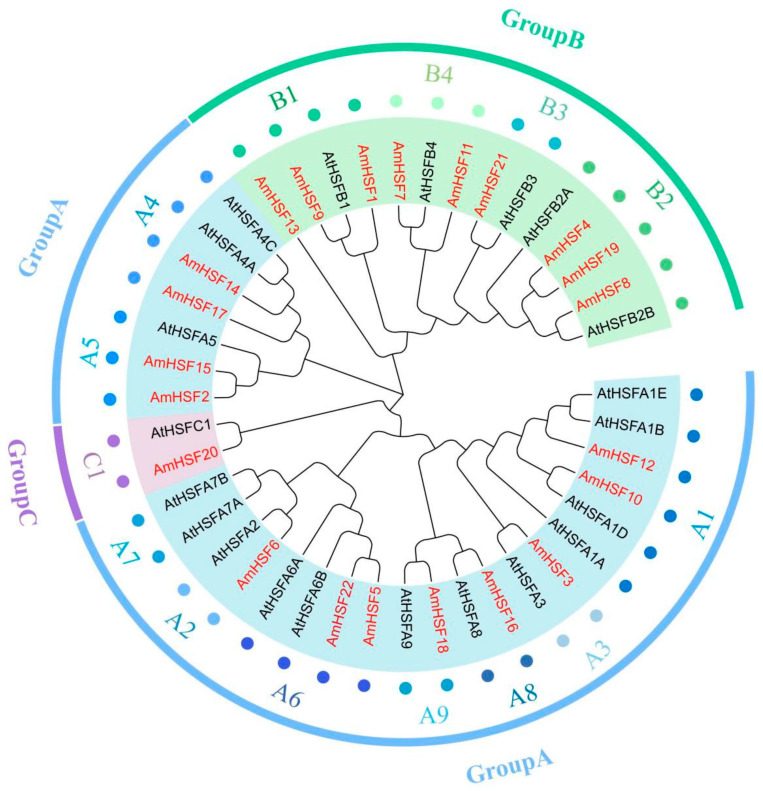
Phylogenetic and classification analysis of HSF proteins in *A. thaliana* and *A. mongholicus.* Different colors indicate the various subgroups. AmHSF protein is highlighted in red.

**Figure 3 biology-13-00280-f003:**
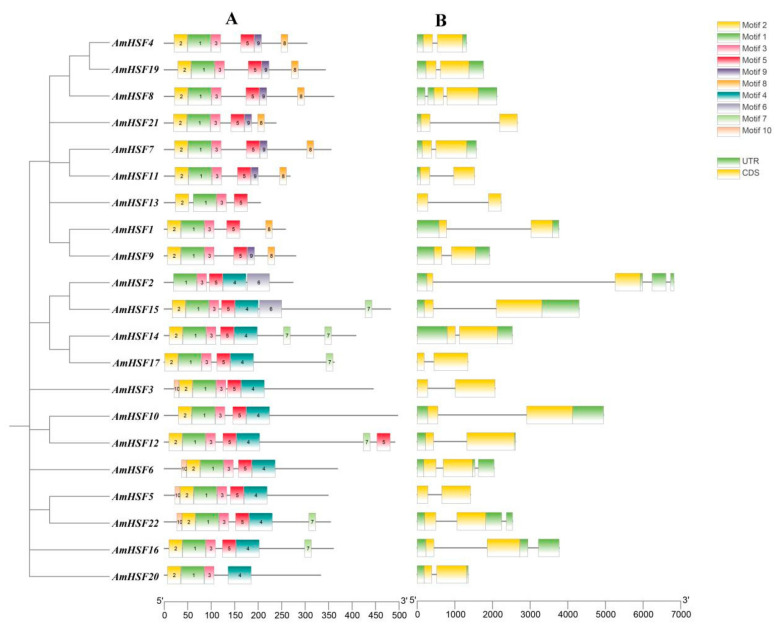
Motif and gene structure features of *AmHSF* genes. (**A**) Distribution of motifs within each AmHSF protein. (**B**) *AmHSF* gene structures. Lines indicate introns.

**Figure 4 biology-13-00280-f004:**
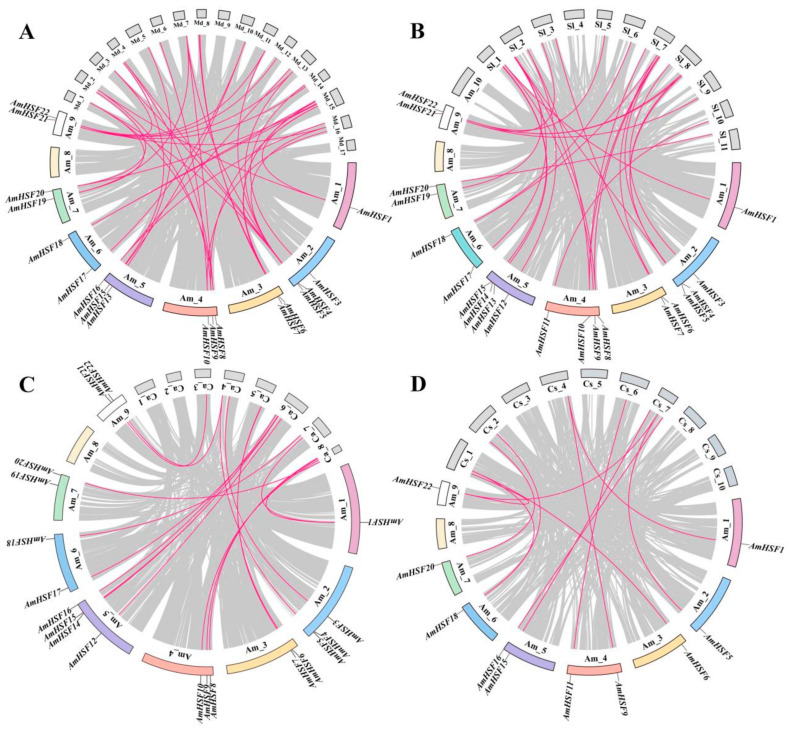
Syntenic analysis of genes between *A. mongholicus* and four other species. (**A**) *M. domestica,* (**B**) *S. lycopersicum*, (**C**) *C. sativa*, (**D**) *C. arietinum*. Red lines represent *AmHSF* genes in homologous pairs. Gray lines symbolize the colinear blocks within *A. mongholicus* and other genomes.

**Figure 5 biology-13-00280-f005:**
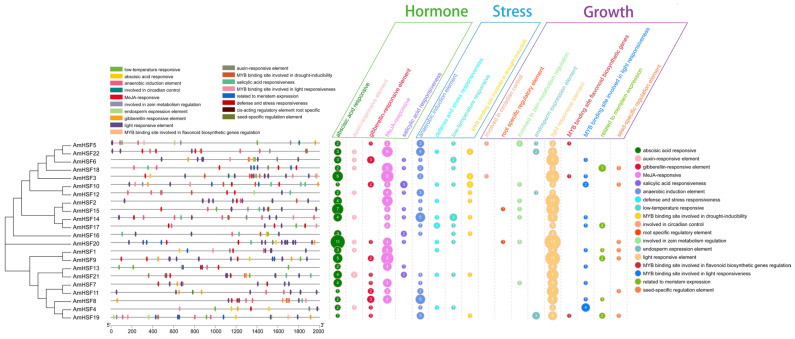
Prediction of *cis*-elements in the promoter regions of 22 AmHSF genes. Different colors represent various *cis*-elements.

**Figure 6 biology-13-00280-f006:**
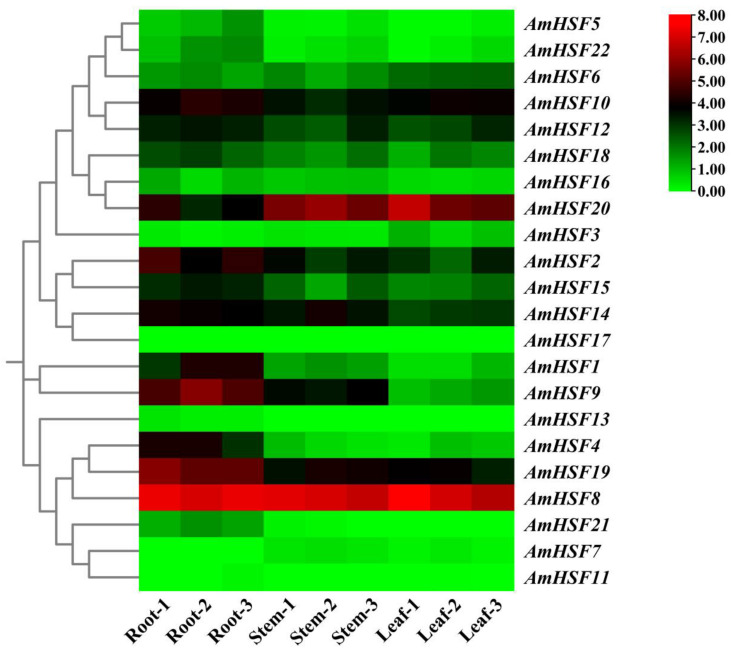
Expression profile analysis of *AmHSF* genes in different tissues of *A. mongholicus*. The color bar on the right side of the figure shows log_2_ FPKM values.

**Figure 7 biology-13-00280-f007:**
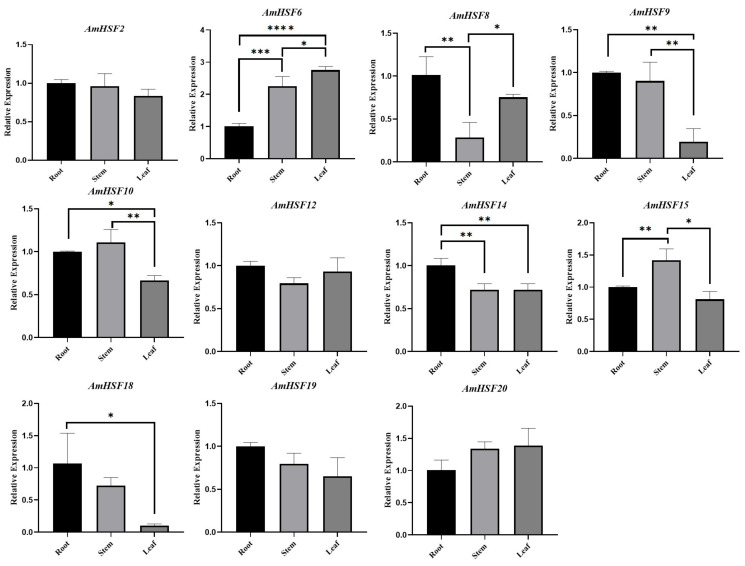
RT-PCR analysis of *AmHSF* genes in different tissues of *A. mongholicus*. Error bars represent three biological replicates and technical replicates. Significance analysis was performed for each component using the one-way ANOVA method. * *p* < 0.05, ** *p* < 0.01, *** *p* < 0.001, and **** *p* < 0.0001.

**Figure 8 biology-13-00280-f008:**
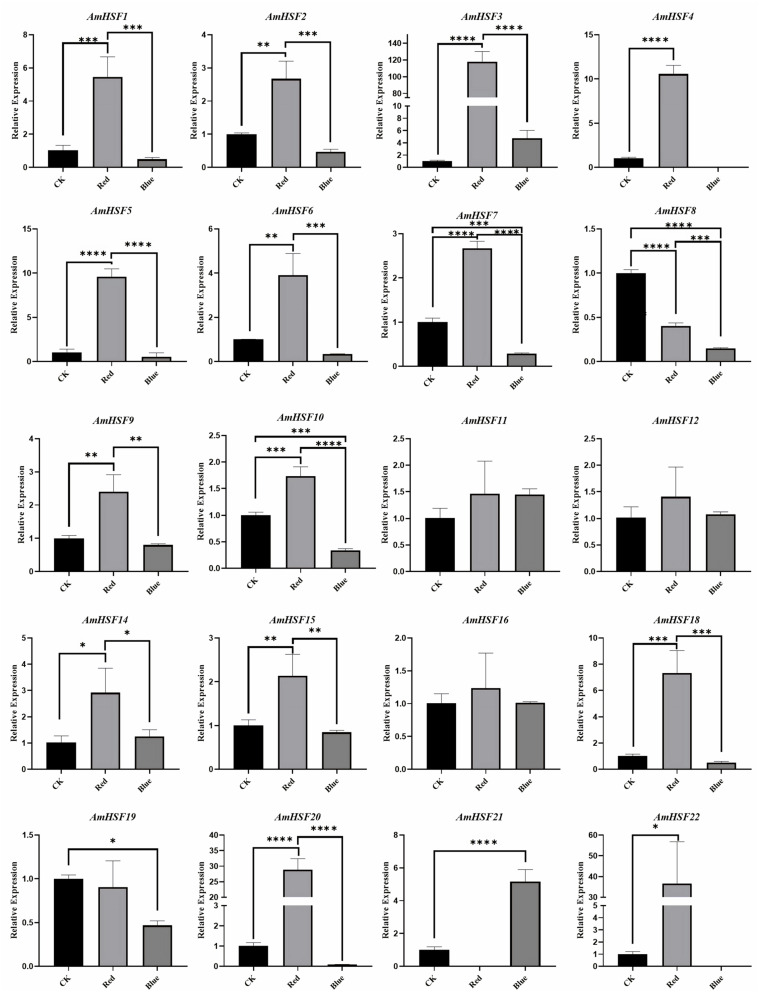
*AmHSF* gene expression levels after different light treatments. Error bars represent three biological replicates and technical replicates. Significance analysis was performed for each component using the one-way ANOVA method. * *p* < 0.05, ** *p* < 0.01, *** *p* < 0.001, and **** *p* < 0.0001.

## Data Availability

The transcriptome data were deposited at NCBI database under accession number PRJNA1064679.

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
