# Peer review of "Genome-Wide Analysis of the HSF Gene Family Reveals Its Role in Astragalus mongholicus under Different Light Conditions"

_biology, 2024, doi:10.3390/biology13040280_

Round 1
Reviewer 1 Report
Comments and Suggestions for Authors
The following comments on the manuscript:
1. Each chromosome containing the AmHSF gene had a different 204
number of genes, except for chromosome 8, which showed a distribution of AmHSF genes. 205
There is a typo in this sentence; chromosome 8 does not contain AmHSF genes.
2.. RT-PCR analysis Statistical processing of data was performed incorrectly. Student's t test is only intended to compare two independent groups when the necessary conditions for the use of parametric criteria. There are three groups in your work: Root, stem, leaf; control, blue light, red light. Comparison three or more independent groups quantitative data is carried out using one-dimensional (one-factor) disperson analysis (One-Way ANOVA) or Kruskal-Wallis test (Kruskal-Wallis test). Kruskal-Wallis test will help you find out if there are differences between groups, but will not be able to show between which groups these differences exist. When statistically significant differences are found between groups using the Kruskal-Wallis test further a posteriori comparison should be made using the Mann-Whitney test. After correcting the manuscript can be recommended for publication in the journal.With respect,
Author Response
Dear Editor and Reviewers :
On behalf of my co-authors, we thank you very much for giving us an opportunity to revise our manuscript. I appreciate your favorable consideration of our manuscript (biology-2970256) and the reviewers’ insightful comments. I have now revised the manuscript according to the reviewers’ comments, which were very helpful. I hope these revisions make the paper acceptable for publication. The reviewers’ comments are addressed point-by-point below, and we have added the "Simple Summary" section in "Abstract".
- Each chromosome containing the AmHSF gene had a different number of genes, except for chromosome 8, which showed a distribution of AmHSF genes. There is a typo in this sentence; chromosome 8 does not contain AmHSF genes.
Responds: I apologize to you for our carelessness. We have modified the expression of this sentence. “Except for chromosome 8, each chromosome contains a different number of AmHSF genes”.
- RT-PCR analysis Statistical processing of data was performed incorrectly. Student's t test is only intended to compare two independent groups when the necessary conditions for the use of parametric criteria. There are three groups in your work: Root, stem, leaf; control, blue light, red light. Comparison three or more independent groups quantitative data is carried out using one-dimensional (one-factor) disperson analysis (One-Way ANOVA) or Kruskal-Wallis test (Kruskal-Wallis test). Kruskal-Wallis test will help you find out if there are differences between groups, but will not be able to show between which groups these differences exist. When statistically significant differences are found between groups using the Kruskal-Wallis test further a posteriori comparison should be made using the Mann-Whitney test. After correcting the manuscript can be recommended for publication in the journal.
Responds: First of all, thank you very much for teaching us so many statistical methods and knowledge, which has laid a solid foundation for our future scientific research. Secondly, according to your suggestion, we used the method of One-Way ANOVA to re-count the data of qPR-PCR and re-describe the data results. Finally, thank you for your valuable suggestions.
With respect,Overall, I found the reviewers’ comments helpful and have revised my paper point-by-point; revisions are marked in red. Thanks again to both the editor and reviewers for their help.
Reviewer 2 Report
Comments and Suggestions for Authors
Light intensity is certainly one of the key abiotic factors affecting plant growth and development.
The authors did a great job; they analyzed the expression of 22 genes from the HSF family in response to a decrease in the intensity of rays of two spectra in the plant Astragalus mongholicus. For this plant species, such work has not been previously carried out, so this study is not only relevant, but also novel.
1) In methodological section there is the sentence about isoflavone detirmination: "Ten biological replicates were collected from each group, rapidly frozen in liquid nitrogen 123 and stored at -80 ℃ for subsequent molecular assays, with the remainder dried at 60 ℃ 124 for isoflavone content determination". But there is no information about the method. Furthermore, there is a paragraph 3.8. Determination of Isoflavone Metabolites and Analysis of AmHSF Expression Patterns after 312 Different Light Treatments. But there is no any data about isoflavone content in 3.8.
2) The title of the MS is "Genome-wide analysis of the HSF gene family reveals its roles in plant development and different light conditions in Astragalus mongholicus". However, the article only contains information about changes in gene expression. I did not find any data on how exactly this affects development (which is stated in the title).
Author Response
Dear Editor and Reviewers :
On behalf of my co-authors, we thank you very much for giving us an opportunity to revise our manuscript. I appreciate your favorable consideration of our manuscript (biology-2970256) and the reviewers’ insightful comments. I have now revised the manuscript according to the reviewers’ comments, which were very helpful. I hope these revisions make the paper acceptable for publication. The reviewers’ comments are addressed point-by-point below, and we have added the "Simple Summary" section in "Abstract".
1. In methodological section there is the sentence about isoflavone detirmination: "Ten biological replicates were collected from each group, rapidly frozen in liquid nitrogen 123 and stored at -80 ℃ for subsequent molecular assays, with the remainder dried at 60 ℃ 124 for isoflavone content determination". But there is no information about the method. Furthermore, there is a paragraph 3.8. Determination of Isoflavone Metabolites and Analysis of AmHSF Expression Patterns after 312 Different Light Treatments. But there is no any data about isoflavone content in 3.8.
Responds: I apologize to you for our carelessness. This part is not included in this manuscript, and we have deleted the relevant content. Again, we apologize for our carelessness.
2. The title of the MS is "Genome-wide analysis of the HSF gene family reveals its roles in plant development and different light conditions in Astragalus mongholicus". However, the article only contains information about changes in gene expression. I did not find any data on how exactly this affects development (which is stated in the title).
Responds: Thank you for your valuable comments. Our idea is to clarify the expression pattern of AmHSF gene in different tissues by using transcriptome data from different tissue sites of Astragalus mongholicus seedlings in soilless cultivation combined with qRT-PCR experiments, and to explore the potential role of AmHSF gene in different tissues by combining bioinformatics methods, so we added the words of plant development in the title. But I realized that this was wrong. Therefore, I deleted plant development and revised the title to show that “Genome-wide analysis of the HSF gene family reveals its role in Astragalus mongholicus under different light conditions”
Overall, I found the reviewers’ comments helpful and have revised my paper point-by-point; revisions are marked in red. Thanks again to both the editor and reviewers for their help.
Round 2
Reviewer 2 Report
Comments and Suggestions for Authors
Dear authors!
Since you decided to exclude data on isoflavones from the article, you probably need to remove or change the sentence on lines 407-410.
Author Response
Dear Editor and Reviewers :
On behalf of my co-authors, we thank you very much for giving us an opportunity to revise our manuscript. I appreciate your favorable consideration of our manuscript (biology-2970256) and the reviewers’ insightful comments. I have now revised the manuscript according to the reviewers’ comments, which were very helpful. I hope these revisions make the paper acceptable for publication. The reviewers’ comments are addressed point-by-point below.
Response to the reviewer’s comments:
- Since you decided to exclude data on isoflavones from the article, you probably need to remove or change the sentence on lines 407-410.
Responds: Thank you for your valuable feedback. Your feedback has made this article more scientific and informative. In response to your question, I would like to make the following response, I hope my response can satisfy you. This section discusses the potential functions of HSF genes under different light conditions. There are references indicating that the HSF gene in Cannabis sativa is positively regulated by red and blue light, and may have the potential to regulate the biosynthesis of cannabinoids. In Astragalus mongholicus, studies have also shown that red and blue light can positively regulate the biosynthesis of isoflavones. Combined with the references and the results of this study, we speculate that the AmHSF gene may be involved in the isoflavone biosynthesis of A. mongholicus under red and blue light, but its true biological function needs to be further verified. We have reorganized this section to make our description more accurate.